



# 1 Evaluation and Calibration of a Low-cost Particle Sensor in Ambient
# 2 Conditions Using Machine Learning Technologies

Minxing Si[1,2,a], Xiong Ying[1,a], Shan Du[3], Ke Du[1*]
[1]Department of Mechanical and Manufacturing Engineering, University of Calgary, 2500 University Drive. NW, Calgary,
AB, Canada, T2N 1N4
[2]Tetra Tech Canada Inc., 140 Quarry Park Blvd, Calgary AB Canada, T2C 3G3
[3]Department of Computer Science, Lakehead University, 955 Oliver Road, Thunder Bay, ON, Canada, P7B 5E1
[*]*Correspondence to* Ke Du (kddu@ucalgary.ca)
[a.] The authors contributed equally to the work.
**Abstract.** Particle sensing technology has shown great potential for monitoring particulate matter (PM) with very few
temporal and spatial restrictions because of low-cost, compact size, and easy operation. However, the performance of low-
cost sensors for PM monitoring in ambient conditions has not been thoroughly evaluated. Monitoring results by low-cost
sensors are often questionable. In this study, a low-cost fine particle monitor (Plantower PMS 5003) was co-located with a
reference instrument, named Synchronized Hybrid Ambient Real-time Particulate (SHARP) monitor, in Calgary Varsity air
monitoring station from December 2018 to April 2019. The study evaluated the performance of this low-cost PM sensor in
ambient conditions and calibrated its readings using simple linear regression (SLR), multiple linear regression (MLR), and
two more powerful machine learning algorithms using random search techniques for the best model architectures. The two
machine learning algorithms are XGBoost and feedforward neural network (NN). Field evaluation showed that the Pearson r
between the low-cost sensor and the SHAPR instrument was 0.78. Fligner and Killeen (F-K) test indicated a statistically
significant difference between the variances of the $PM_{2.5}$ values by the low-cost sensor and by the SHARP instrument. Large
overestimations by the low-cost sensor before calibration were observed in the field and were believed to be caused by the
variation of ambient relative humidity. The root mean square error (RMSE) was 9.93 when comparing the low-cost sensor
with the SHARP instrument. The calibration by the feedforward NN had the smallest RMSE of 3.91 in the test dataset,
compared to the calibrations by SLR (4.91), MLR (4.65), and XGBoost (4.19). After calibrations, the F-K test using the test
dataset showed that the variances of the $PM_{2.5}$ values by the NN and the XGBoost and by the reference method were not
statistically significantly different. From this study, we conclude that feedforward NN is a promising method to address the
poor performance of the low-cost sensors for $PM_{2.5}$ monitoring. In addition, the random search method for hyperparameters
was demonstrated to be an efficient approach for selecting the best model structure.
**Keywords:** Low-cost sensor, machine learning, TensorFlow, XGBoost, $PM_{2.5}$



## 1 Introduction

Particular matter (PM), whether it is natural or anthropogenic, has pronounced effects on human health, visibility, and global climate (Charlson et al., 1992; Seinfeld and Pandis, 1998). To minimize the harmful effects of PM pollution, the Government of Canada launched the National Air Pollution Surveillance (NAPS) program in 1969 to monitor and regulate PM and other criteria air pollutants in populated regions, including ozone ($O_3$), sulfur dioxide ($SO_2$), carbon monoxide (CO), nitrogen dioxide ($NO_2$). Currently, PM monitoring is routinely carried out at 286 designated air sampling stations in 203 communities in all provinces and territories of Canada (Government of Canada, 2019). Many of the monitoring stations use Beta Attenuation Monitor (BAM), which is based on the adsorption of beta radiation, or Tapered Element Oscillating Microbalance (TEOM) instrument, which is a mass-based technology to measure PM concentrations. An instrument that combines two or more technologies, such as Synchronized Hybrid Ambient Real-time (SHARP), is also used in some monitoring stations. The SHARP instrument combines light scattering with beta attenuation technologies to determine PM concentrations.

Although these instruments are believed to be accurate for measuring PM concentration and have been widely used by many air monitoring stations worldwide (Chow and Watson, 1998; Patashnick and Rupprecht, 1991), they have common drawbacks: they can be challenging to operate, bulky, and expensive. The instrument costs from 8,000 Canadian dollars (CAD) to tens of thousands of dollars (Chong and Kumar, 2003). The SHARP instrument used in this study as a reference method costs approximately $40,000 (CDNova Instrument Ltd., 2017). Significant resources, such as specialized personnel or technicians, are also required for regular system calibration and maintenance. In addition, the sparsely spread stations may only represent PM levels in limited areas near the stations because PM concentrations vary spatially and temporally depending on local emission sources as well as meteorological conditions (Xiong et al., 2017). Such a low-resolution PM monitoring network cannot support public exposure and health effects studies that are related to PM, because these studies require high spatial- and temporal-resolution of monitoring network in the community (Snyder et al., 2013). In addition, the well-characterized scientific PM monitors are not portable due to their large size and volumetric flow rate, which means they are not practical for measuring personal PM exposure (White et al., 2012).

As a possible solution to the above problems, a large number of low-cost PM sensors could be deployed, and a high-resolution PM monitoring network could be constructed. Low-cost PM sensors are portable and commercially available. They are cost-effective and easy to deploy, operate, and maintain, which offers significant advantages compared to conventional analytical instruments. If many low-cost sensors are deployed, PM concentrations can be monitored continuously and simultaneously at multiple locations for a reasonable cost (Holstius et al., 2014). A dense monitoring network using low-cost sensors can also assist in mapping hotspots of air pollution, creating emission inventories of air pollutants, and estimating adverse health effects due to personal exposure to the PM (Kumar et al., 2015).



However, low-cost sensors present challenges for broad application and installation. Most sensor systems have not been
thoroughly evaluated (Williams et al., 2014), and the data generated by these sensors are of questionable quality (Wang et
al., 2015). Currently, most low-cost sensors are based on laser light scattering technology (LLS), and the accuracy of LLS is
mostly affected by particle composition, size distribution, shape, temperature, and relative humidity (Jayaratne et al., 2018;
Wang et al., 2015).
Several studies evaluated LLS sensors by comparing the performance of low-cost sensors with medium- to high-cost
instruments under laboratory and ambient conditions. For example, Zikova et al. (2017) used low-cost Speck monitors to
measure $PM_{2.5}$ concentrations in indoor and outdoor environments, and the low-cost sensors overestimated the concentration
by 200% for indoor and 500% for outdoor, compared to a reference instrument – Grimm 1.109 dust monitor. Jayaratne et al.
(2018) reported that $PM_{10}$ concentrations generated by a Plantower low-cost particle sensor (PMS 1003) were 46% greater
than a TSI 8350 DustTrak DRX aerosol monitor under a foggy environment. Wang et al. (2015) compared PM
measurements from three low-cost LLS sensors – Shinyei PPD42NS, Samyoung DSM501A, and Sharp GP2Y1010AU0F –
with a SidePack (TSI Inc.) using smoke from burning incense. High linearity was found with $R^2$ greater than 0.89, but the
responses depended on particle composition, size, and humidity. Air Quality Sensor Performance Evaluation Center (AQ-
SPEC) of South Coast Air Quality Management District (SCAQMD) also evaluated the performances of three Purple Air
PA-II sensors (model: Plantower PMS 5003) by comparing their readings with two United States Environmental Protection
Agency (US EPA) Federal Equivalent Method (FEM) instruments – BAM (MetOne) and Grimm dust monitors in laboratory
and field environments in south California (Papapostolou et al., 2017). Overall, the three sensors showed moderate to good
accuracy, compared to the reference instrument for $PM_{2.5}$ for a concentration range between 0 to 250 µg m$^{-3}$. Lewis et al.
(2016) evaluated low-cost sensors in the field for $O_3$, nitrogen oxide (NO), $NO_2$, volatile organic compounds (VOCs), $PM_{2.5}$,
and $PM_{10}$; only $O_3$ sensors showed good performance compared to the reference measurements.
Several studies developed calibration models using multiple techniques to improve low-cost sensors' performance. For
example, De Vito et al. (2008) tested feedforward neural network (NN) calibration for benzene monitoring and reported a
further calibration was needed for low concentrations. Bayesian optimization was also used to search feedforward NN
structures for the calibrations of CO, $NO_2$, and $NO_x$ low-cost sensors (De Vito et al., 2009). Zheng et al. (2018) calibrated
Plantower low-cost particle sensor PMS 3003 by fitting a linear least-squares regression model. A nonlinear response was
observed when ambient $PM_{2.5}$ exceeded 125 ug m$^{-3}$. The study concluded that a quadratic fit was more appropriate than a
linear model to capture this nonlinearity.
Zimmerman et al. (2018) explored three different calibration models, including laboratory univariate linear regression,
empirical MLR, and a more modern machine learning algorithm, random forests (RF), to improve Real-time Affordable
Multiple-Pollutant (RAMP) sensor's performance. They found that the sensors calibrated by RF models improved their
accuracy and precision over time, with average relative errors of 14% for CO, 2% for $CO_2$, 29% for $NO_2$, and 15% for $O_3$.


The study concluded that combing RF models with low-cost sensors is a promising approach to address the poor
performance of low-cost air quality sensors.
Spinelle et al. (2015) reported several calibration methods for low-cost $O_3$ and $NO_2$ sensors. The best calibration method
for $NO_2$ was an NN algorithm with feedforward architecture. $O_3$ could be calibrated by simple linear regression (SLR).
Spinelle et al. (2017) also evaluated and calibrated NO, CO, and $CO_2$ sensors, and the calibrations by feedforward NN
architectures showed the best results. Similarly, Cordero et al. (2018) performed a two-step calibration for an AQmesh $NO_2$
sensor using supervised machine learning regression algorithms, including NNs, RFs, and Support Vector Machines
(SVMs). The first step produced an explanatory variable using multivariate linear regression. In the second step, the
explanatory variable was fed into machine learning algorithms, including RF, SVM, and NN. After the calibration, the
AQmesh $NO_2$ sensor met the standards of accuracy for high concentrations of $NO_2$ in the European Union's Directive
2008/50/EC on Air Quality. They highlighted the need to develop an advanced calibration model, especially for each sensor,
as the responses of individual sensors are unique.
Williams et al. (2014) evaluated eight low-cost PM sensors; the study showed frequent disagreement between the low-
cost PM sensors and FEMs. In addition, the study concluded that the performances of the low-cost sensors were significantly
impacted by temperature and relative humidity (RH). Recurrent NN architectures were also tested for the calibrations of
some gas sensors (De Vito et al., 2018; Esposito et al., 2016). The results showed that the dynamic approaches performed
better than traditional static calibration approaches. Calibrations of $PM_{2.5}$ sensors were also reported in recent studies. Lin et
al. (2018) performed two-step calibrations for $PM_{2.5}$ sensors using 236 hourly data collected on buses and road cleaning
vehicles. The first step was to construct a linear model, and the second step used RF machine learning for further calibration.
The RMSE after the calibrations was 14.76 µg m$^{-3}$, compared to a reference method. The reference method used in this study
was a Dylos DCI1700 device, which is not a US EPA federal reference method (FRM) or FEM. Loh and Choi (2019) trained
and tested SVC, k-nearest neighbor, RF, and XGBoost machine learning algorithms to calibrate $PM_{2.5}$ sensors using 319
hourly data. XGBoost archived the best performance with a RMSE of 5.0 µg m$^{-3}$. However, the low-cost sensors in this
study were not co-located with the reference method, and the machine learning models were not tested using unseen data
(test data) for predictive power and overfitting.
Although there are studies in calibrating low-cost sensors, most of them focused on gas sensors or used short-term data to
calibrate PM sensors. To our best knowledge, no one has reported studies on PM sensor calibration using random search
techniques for the best machine learning model's configuration under ambient conditions during different seasons. In this
study, a low-cost fine particle monitor (Plantower PMS 5003) was co-located with a SHARP monitor Model 5030 at Calgary
Varsity Air Monitoring Station in an outdoor environment from December 7, 2018, to April 26, 2019. The SHARP
instrument is the reference method in this study and is a US EPA FEM (US EPA, 2016). The objectives of this stuudy are:
(1) to evaluate the performance of the low-cost PM sensor in a range of outdoor environmental conditions by comparing its



PM$_{2.5}$ readings with those obtained from the SHARP instrument; and (2) to assess four calibration methods: a) a SLR or
univariate linear regression based on the low-cost sensor values; b) a multiple linear regression (MLR) using the PM$_{2.5}$, RH,
and temperature measured by the low-cost sensor as predictors; c) a decision-tree-based ensemble algorithm, called
XGBoost or Extreme Gradient Boosting; and d) a feedforward NN architecture with a backpropagation algorithm.

XGBoost and NN are the most popular algorithms used on Kaggle – a platform for data science and machine learning

competition. In 2015, 17 winners out of 29 competitions on Kaggle used XGBoost, 11 winners used deep NN algorithm
(Chen and Guestrin, 2016).

This study is unique in the following ways:

1)   To the best of our knowledge, this is the first comprehensive study using long-term data to calibrate low-cost

particle sensors in the field. Most previous studies focused on calibrating gas sensors (Maag et al., 2018). There are

two studies on PM sensor calibrations using machine learning, but they used a short-term dataset that did not

include seasonal changes in ambient conditions (Lin et al., 2018; Loh and Choi, 2019). The shortcomings of the two

studies were discussed above.

2)   Although several studies researched the calibration of gas sensors using NN, this study explores multiple

hyperparameters to search for the best NN architecture. Previous research configured one to three hyperparameters,

compared to six in this study (De Vito et al., 2008, 2009, 2018; Esposito et al., 2016; Spinelle et al., 2015, 2017). In

addition, this study tested the Rectified Linear Unit (ReLU) as the activation function in the feedforward NN.

Compared to sigmoid and tanh activation functions used in the previous studies for NN calibration models, the

ReLU function can accelerate the convergence of stochastic gradient descent to a factor of 6 (Krizhevsky et al.,

2017).

3)   Previous NN and tree-based calibration models used manual search or grid search for hyperparameters tuning. This

study introduced random search method for the best calibration models. Random search is more efficient than

traditional manual and grid search (Bergstra and Bengio, 2012) and evaluates more of the search space, especially

when search space is more than three dimensions (Timbers, 2017).

## 2 Method

### 2.1 Data preparation

One low-cost sensor unit was provided by Calgary-based company SensorUp and deployed at the Varsity station in the
Calgary Reginal Airshed Zone (CRAZ) in Calgary, Alberta, Canada. The unit contains one sensor, one electrical board, and
one housing as a shelter. The sensor in the unit is Plantower PMS 5003, and it measured outdoor fine particle (PM$_{2.5}$)


concentrations (µg m⁻³), air temperature (°C), and RH (%) every six seconds. The minimum detectable particle diameter by
the sensor is 0.3 µm. The instrument costs approximately $20 CAD and is referred to as the low-cost sensor in this paper.
The low-cost sensor is based on LLS technology; PM$_{2.5}$ mass concentration is estimated from the detected amount of
scattered light. The LLS sensor is installed on the electrical board and then placed in the shelter for outdoor monitoring. The
unit has a wireless link to a router in the Varsity station. A picture of the low-cost sensor and the monitoring environment
where the low-cost sensor unit and the SHARP instrument were co-located is provided in Fig. 1. The router uses cellular
service to transfer the data from the low-cost sensor to SensorUp's cloud data storage system. The measured outdoor PM$_{2.5}$,
temperature, and RH data at a six-second interval from 00:00 on December 7, 2018, to 23:00 on April 26, 2019, were
downloaded from the cloud data storage system for evaluation and calibration.

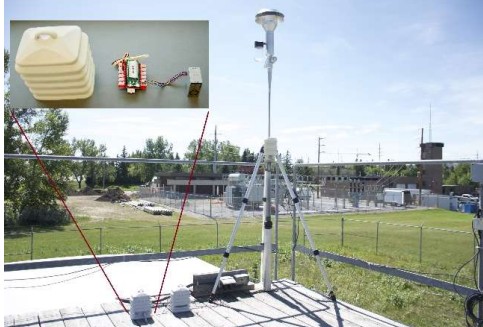

**Figure 1:** The low-cost sensor used in the study and the ambient inlet of the reference method – SHARP Model 5030
The reference instrument used to evaluate the low-cost sensor is a Thermal Fisher Scientific's SHARP Model 5030. The
SHARP instrument was installed at the Calgary Varsity station by CRAZ. The SHARP instrument continuously uses two
compatible technologies, light scattering and beta attenuation, to measure PM$_{2.5}$ every six minutes with an accuracy of ±5%.
The SHARP instrument is operated and maintained by CRAZ in accordance with the provincial government's guideline
outlined in Alberta's air monitoring directive. Hourly PM$_{2.5}$ data are published on the Alberta Air Data Warehouse website
(http://www.airdata.alberta.ca/). The Calgary Varsity station also continuously monitors CO, methane, oxides of nitrogen,
non-methane hydrocarbons, outdoor air temperature, O$_3$, RH, total hydrocarbon, wind direction, and wind speed. Detailed
information on the analytical systems for the CRAZ Varsity station can be found on their website
(https://craz.ca/monitoring/info-calgary-nw/).
The ambient conditions in this study measured by the SHARP instrument are presented in Table 1.



**Table 1: Ambient Condition Measured by SHARP**

| Climate Data | SHARP Value |
|---|---|
| Temperature | -31.4 ℃ ~ 19 ℃ |
| RH | 10% ~ 99% |
| Wind Speed | 4.3 ~ 37.1 km/h 10 m |


The following steps were taken to process the raw data from 00:00 on December 7, 2018, to 23:00 on April 26, 2019:
1)   The six-second interval data recorded by the low-cost sensor, including $PM_{2.5}$, temperature, and RH, were averaged
181         into hourly data to pair with SHARP data because only hourly SHARP data are publicly available.

2)   The hourly sensor data and hourly SHARP data were combined into one structured data table. $PM_{2.5}$, temperature,
183         and RH by the low-cost sensor as well as $PM_{2.5}$ by SHARP columns in the data table were selected. The data table
184         then contains 3,384 rows and four columns. Each row represents one hourly data point. The columns include the data
measured by the low-cost sensor and the SHARP instrument.

3)   Rows in the data table with missing values were removed – 299 missing values for $PM_{2.5}$ from the low-cost sensor
and 36 missing values for $PM_{2.5}$ from the SHARP instrument. The reason for missing data from the SHARP
instrument is because of the calibration. However, the reason for missing data from the low-cost sensor is unknown.

4)   The data used for NN were transformed by z standardization with a mean of zero and a standard deviation of one.
After the above steps, the processed data table with 3,050 rows and four columns was used for evaluation and calibration.
The data file is provided in the supplementary information of this paper. Each row represents one example or sample for the
training or testing by the calibration methods.
**2.2 Low-cost sensor evaluation**
Pearson correlation coefficient was used to compare the correlation for $PM_{2.5}$ values between the low-cost sensor and the
SHARP. SHAPR was the reference method. The $PM_{2.5}$ data by the low-cost sensor and SHARP were also compared using
root mean square error (RMSE), mean square error (MSE), and mean absolute error (MAE).
Fligner and Killeen test (F-K test) was used to evaluate the equality (homogeneity) of variances for $PM_{2.5}$ values between
the low-cost sensor and the SHARP instrument (Fligner and Killeen, 1976). F-K test is a superior option in terms of
robustness and power when data are non-normally distributed, the population means are unknown, or outliers cannot be
removed (Conover et al., 1981; de Smith, 2018). The null hypothesis of the F-K test is that all populations' variances are
equal; the alternative hypothesis is that the variances are statistically significantly different.





**2.3 Calibration**
Four calibration methods were evaluated: SLR, MLR, XGBoost, and NN. Some predictions from the SLR, MLR, and
XGBoost have negative values because they extrapolate observed values and regression is unbounded. When the predicted
$PM_{2.5}$ values generated by these calibration methods were negative, the negative values were replaced with the sensor data.
MLR, XGBoost, and feedforward NN use the $PM_{2.5}$, temperature, and RH data measured by the low-cost sensor as
inputs. The $PM_{2.5}$ measured by the SHARP instrument is used as the target to supervise the machine learning process. The
processed dataset with 3,050 rows and four columns was randomly shuffled and then divided into a training set, which was
the data used to build models and minimize the loss function, and a test set, which was the data that the model has never run
with before testing (Si et al., 2019). The test dataset was only used once and gave an unbiased evaluation of the final model's
performance. The evaluation was to test the ability of the machine learning model to provide sensible predictions with new
inputs (LeCun et al., 2015). The training dataset had 2,440 examples (samples). The test dataset had 610 examples (samples).
**2.3.1 Simple linear regression and multiple linear regression**
The calibration by a SLR used Equation 1.
$\hat{y} = \beta_0 + \beta_1 \times PM_{2.5}$ (1)
$\beta_0$ and $\beta_1$ are the model coefficient and were calculated using the training dataset. $\hat{y}$ is model predicted (calibrated) values.
$PM_{2.5}$ is the value measured by the low-cost sensor.
The MLR used $PM_{2.5}$, RH, and temperature measured by the low-cost sensor as predictors because the low-cost sensor
only measured these parameters. The model is expressed as Equation 2.
$\hat{y} = \beta_0 + \beta_1 \times PM_{2.5} + \beta_2 \times T + \beta_3 \times RH$ (2)
The model coefficients, $\beta_0$ to $\beta_3$, were calculated using the training dataset with SHARP provided readings as $\hat{y}$. The
outputs of the models generated by the SLR and MLR were evaluated by comparing to the SHARP's readings in the test
dataset.
**2.3.2 XGBoost**
XGBoost is a scalable decision tree-based ensemble algorithm, and it uses a gradient boosting framework (Chen and
Guestrin, 2016). The XGBoost was implemented using the XGBoost (Version 0.90) and sklearn (Version 0.21.2) packages
in Python (Version 3.7.3). Random search method (Bergstra and Bengio, 2012) was used to tune the hyperparameters in the
XGBoost algorithm, and the hyperparameters tuned include
• Number of trees to fit (n_estimator)
• Maximum depth of a tree (max_depth)
• Step size shrinkage used in update (learning_rate)





• Subsample ratio of columns when constructing each tree (colsample_bytree)
• Minimum loss reduction required to make a further partition on a leaf node of the tree (gamma)
• L2 regularization (Ridge Regression) on weights (reg_lambda)
• Minimum sum of instance weight needed in a child (min_child_weight)
Ten-fold cross-validation was used to select the best model with minimum MSE from the random search. The best model
was then evaluated against the SHARP $PM_{2.5}$ data using the test dataset.

### 2.3.3 Neural network

A fully connected feedforward NN architecture was used in the study. In a fully connected NN, each unit (node) in a
layer is connected to each unit in the following layer. Data from the input layer are passed through the network until the
unit(s) in the output layer is (are) reached. An example of a fully connected feedforward NN is presented in Fig.2. A
backpropagation algorithm is used to minimize the difference between the SHARP measured values and the predicted values
(Rumelhart et al., 1986).

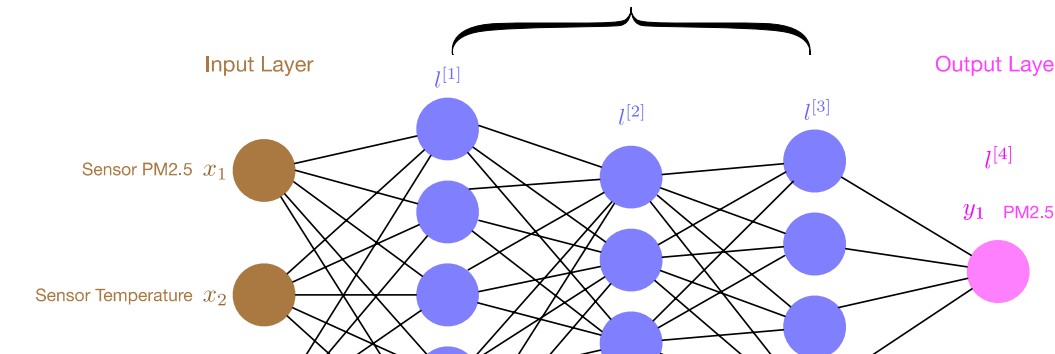

**Figure 2:** Example of a Neural Network Structure
The NN was implemented using the Keras (Version 2.2.4) and TensorFlow (Version 1.14.0) libraries in Python (Version
3.7.3). Keras and TensorFlow were the most referenced deep learning framework in scientific research in 2017 (RStudio,
2018). Keras is the front end of TensorFlow.





Learning rate, L2 regularization rate, numbers of hidden layers, number of units in the hidden layers, and optimization
methods were tuned using random search method provided in the scikit-learn machine learning library. Ten-fold cross-
validation was used to evaluate the models. The model with the minimum MSE was considered to be the best-fit model and
then used for model testing.
**3 Results and Discussion**
**3.1 Sensor evaluation**
**3.1.1 Hourly data**
The RMSE, MSE, and MAE between the low-cost sensor and SHARP for the hourly $PM_{2.5}$ data were 10.58, 111.83, and
5.74. The Pearson correlation coefficient r value was 0.78. The $PM_{2.5}$ concentrations by the sensor ranged from 0 µg m$^{-3}$ to
178 µg m$^{-3}$ with a standard deviation of 14.90 µg m$^{-3}$ and a mean of 9.855 µg m$^{-3}$. The $PM_{2.5}$ concentrations by SHARP
ranged from 0 µg m$^{-3}$ to 80 µg m$^{-3}$ with a standard deviation of 7.80 and a mean of 6.55 µg m$^{-3}$. Both SHARP and the low-
cost sensor dataset had a median of 4.00 µg m$^{-3}$ based on hourly data (Fig.3). The p-value from the F-K test was less than
$2.2 \times 10^{-16}$, indicating that the variance of the $PM_{2.5}$ values measured by the low-cost sensor was statistically significantly
different from the variance of the $PM_{2.5}$ values measured by the SHARP instrument.






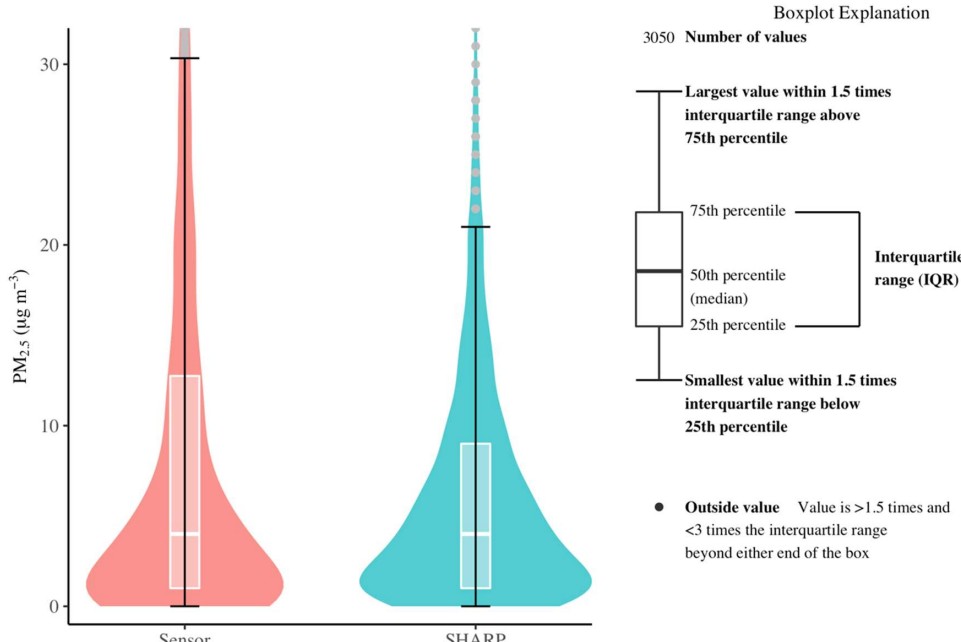


**Figure 3:** Comparison of the Hourly PM$_{2.5}$ Values between the Low-Cost PM Sensor and SHARP**.** Based on 3,050 hourly paired data. The low-cost sensor has 250 hourly data greater than 30 µg m$^{-3}$. SHARP has 174 hourly data greater than 20 µg m$^{-3}$. Bars indicate the 25th and 75th percentile values, whiskers extend to values within 1.5 times IQR, and dots represent values outside of the IQR. The boxplot explanation on the right is adjusted from DeCicco (2016)

**3.1.2 24 Hour rolling average data**

Over 24 hours, the median value for SHARP was 5.38 µg m$^{-3}$ and for the low-cost sensor was 5.01 µg m$^{-3}$. Over five months (December 2018 to April 2019), the low-cost sensor tended to generate higher PM$_{2.5}$ values compared to the SHARP monitoring data (Fig. 4)




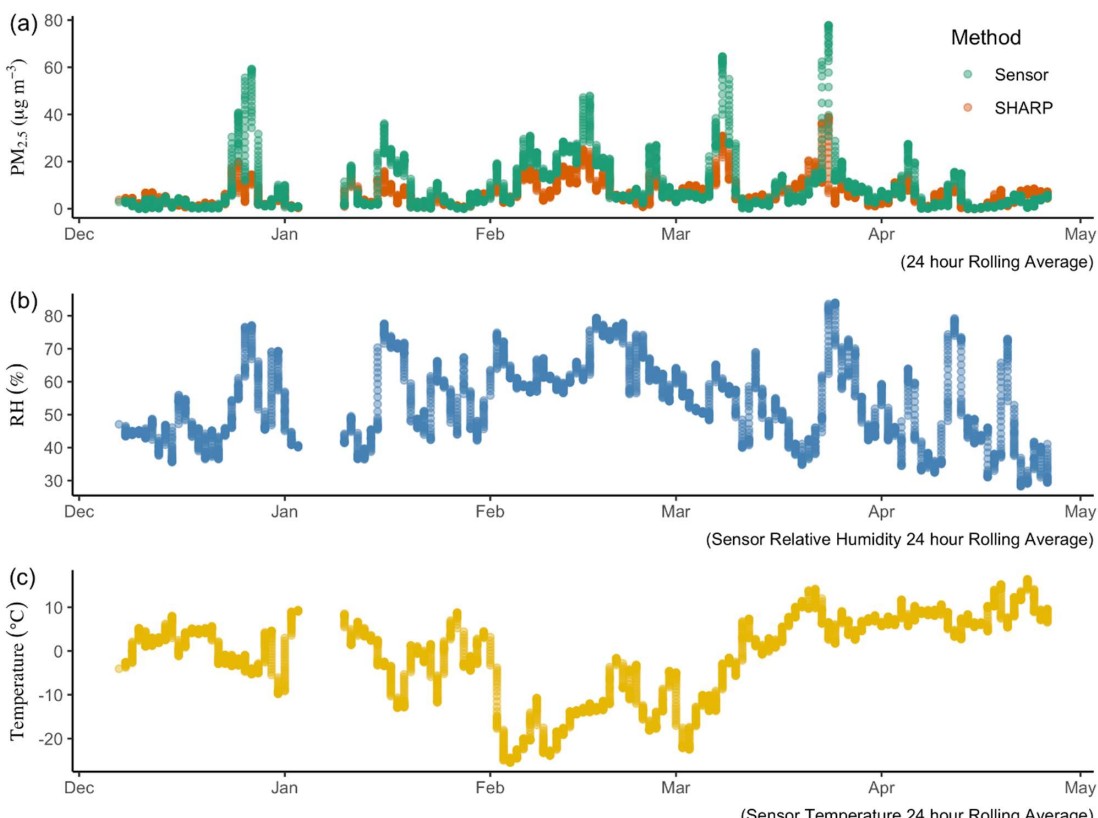


**Figure 4:** PM2.5, Relative Humidity, and Temperature data on the basis of 24 hour rolling average

When PM$_{2.5}$ concentrations were greater than 10 µg m$^{-3}$, the low-cost sensor consistently produced values that were
higher than the reference method (Fig.5). When the concentrations were less than 10 µg m$^{-3}$, the performance of the low-cost
sensor was closed to the reference method producing slightly smaller values (Fig. 5)





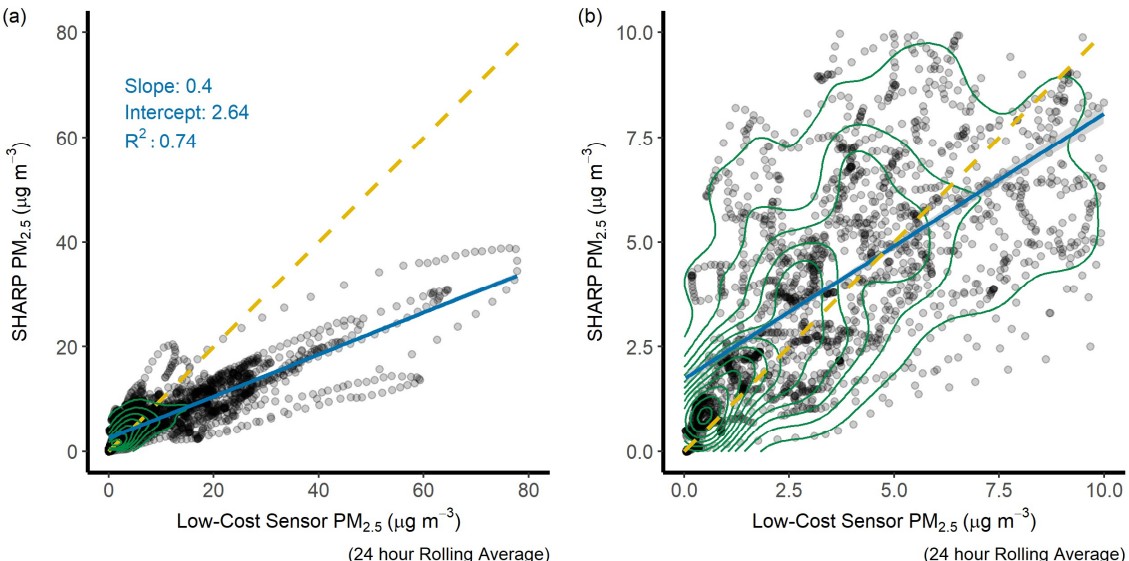


**Figure 5:** SHARP verse Low-Cost Sensor $PM_{2.5}$ Concentration ($\mu g\ m^{-3}$). The yellow dashed line is a 1:1 line. The solid blue line is a
regression line. (a) plot is in full scale, (b) plot is a zoom-in plot of plot a. The green circle represents data density.

**3.2 Calibration by simple linear regression and multiple linear regression**
The RMSE was 4.91 calibrated by SLR and 4.65 by MLR (Table 2). The r value was 0.74 by the SLR and 0.77 by MLR .
The p-values in the F-K test by the SLR and MRL were less than 0.05, which suggested that the variances of the $PM_{2.5}$
values were statistically significantly different.
**Table 2: Calibration Results by SLR and MLR using Test Dataset**

| Criteria | Low-Cost Sensor | SLR | MLR |
|---|---|---|---|
| RMSE | 9.93 | 4.91 | 4.65 |
| MSE | 98.62 | 24.09 | 21.61 |
| MAE | 5.63 | 3.21 | 3.09 |
| Pearson r | 0.74 | 0.74 | 0.77 |
| p-value in the F-K test | $7.062 \times 10^{-09}$ | $5.81 \times 10^{-13}$ | $9.90 \times 10^{-10}$ |
| $\beta_0$ | - | 2.49 | 8.47 |
| $\beta_1$ | | 0.41 | 0.46 |
| $\beta_2$ | | | -0.12 |
| $\beta_3$ | | | -0.0055 |

Note: The test dataset contains 660 examples.



**3.3 Calibration by XGBoost**

The hyperparameters selected by the random search for the best model using XGBoost is presented in Table 3.

**Table 3: Hyperparameters for the Best XGBoost Model**

| XGBoost Hyperparameters | Values |
|---|---|
| Number of trees to fit (n_estimator) | 37 |
| Maximum depth of a tree (max_depth) | 9 |
| Step size shrinkage used in update (learning_rate) | 0.33 |
| Subsample ratio of columns when constructing each tree (colsample_bytree) | 0.83 |
| Minimum loss reduction required to make a further partition on a leaf node of the tree (gamma) | 6.36 |
| L2 regularization (Ridge Regression) on weights (reg_lambda) | 33.08 |
| Minimum sum of instance weight needed in a child (min_child_weight) | 25.53 |

In the training dataset, the RMSE was 3.03, and the MAE was 1.93 by the best XGBoost model. The RMSE in the test dataset reduced by 57.8% using the XGBoost from 9.93 by the sensor to 4.19 (Table 4). The p-value in the F-K test using the test dataset was 0.7256, which showed no evidence that the $PM_{2.5}$ values varied with statistical significance between the XGBoost predicted values and SHARP measured values.

**Table 4: Calibration Results by XGBoost using Test Dataset**

| Criteria | Low-Cost Sensor | XGBoost |
|---|---|---|
| RMSE | 9.93 | 4.19 |
| MSE | 98.62 | 17.61 |
| MAE | 5.63 | 2.63 |
| Pearson r | 0.74 | 0.82 |
| p-value in the F-K test | $7.062 \times 10^{-09}$ | 0.7256 |

Note: The test dataset contains 610 examples.

**3.4 Calibration by neural network**

The hyperparameters for the best NN model are presented in Table 5.

**Table 5: Hyperparameters for the Best Neural Network Model**

| NN Hyperparameters | Values |
|---|---|
| Learning_rate | 0.001 |
| L2 regularization | 0.01 |
| Numbers of hidden layer(s) | 5 |
| Numbers of units in the hidden layer(s) | 32-32-32-32-32 |
| Optimization method | Nadam |





| Epochs | 100 |
| --- | --- |


In the training dataset, the RMSE was 3.22, and the MAE was 2.17 by the best NN-based model. The RMSE reduced by
60% using the NN from 9.93 to 3.91 in the test dataset (Table 6). The p-value in the F-K test was 0.43, which suggested that
the variances in the $PM_{2.5}$ values were not statistically significantly different between the NN predicted values and SHARP
measured values.
**Table 6: Calibration Results by Neural Network using Test Dataset**

| Criteria | Low-Cost Sensor | Neural Network |
| --- | --- | --- |
| RMSE | 9.93 | 3.91 |
| MSE | 98.62 | 15.26 |
| MAE | 5.63 | 2.38 |
| Pearson r | 0.74 | 0.85 |
| p-value in the F-K test | $7.062 \times 10^{-09}$ | 0.43 |

Note: the test dataset includes 610 examples.
**3.5 Discussion**
**3.5.1 Relative humidity impact**
RH has significant effects on the low-cost sensor's responses. The RH trend matched the low-cost sensor's $PM_{2.5}$ trend
closely. The spikes in the low-cost sensor's $PM_{2.5}$ trend corresponded with the increases of RH values, and the low-cost
sensor intended to produce inaccurate high $PM_{2.5}$ values when RH suddenly increased (Fig. 4). However, the relationship
between $PM_{2.5}$ and RH was not linear (Fig. 6)





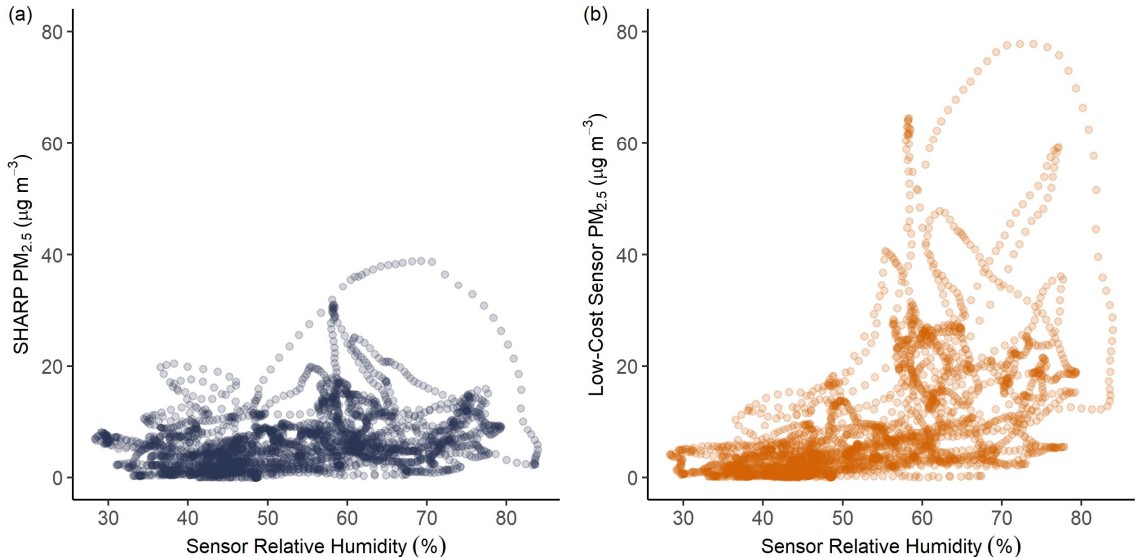


**Figure 6:** PM$_{2.5}$ verse Relative Humidity
**3.5.2 Calibration assessment**
Descriptive statistics of the PM$_{2.5}$ concentrations in the test dataset for SHARP, low-cost sensor, XGBoost, NN, SLR, and
MLR are presented in Table 7. The arithmetic mean of the PM$_{2.5}$ concentrations measured by the low-cost sensor was
9.44 μg m$^{-3}$. In contrast, the means of the PM$_{2.5}$ concentrations were 6.44 μg m$^{-3}$ by SHARP, 6.40 μg m$^{-3}$ by XGBoost, and
6.09 μg m$^{-3}$ by NN.
**Table 7**: Descriptive statistics of PM$_{2.5}$ Concentrations using the Test Dataset

| PM2.5 Concentration (μg m$^{-3}$) | SHARP | Low-Cost Sensor | XGBoost | NN | SLR | MLR |
|---|---|---|---|---|---|---|
| Minimum | 0.00 | 0.00 | 0.00 | 0.19 | 2.49 | 0 |
| 1$^{st}$ quartile | 2.00 | 0.083 | 2.09 | 1.78 | 2.83 | 3.27 |
| Median | 4.00 | 4.00 | 4.98 | 4.16 | 4.13 | 4.79 |
| Mean | 6.44 | 9.44 | 6.40 | 6.09 | 6.37 | 6.42 |
| 3$^{rd}$ quartile | 8.00 | 11.94 | 8.61 | 8.20 | 7.39 | 7.18 |
| Maximum | 49.00 | 103.33 | 39.94 | 47.19 | 44.97 | 48.56 |





| SD | 7.32 | 13.53 | 6.03 | 6.23 | 5.57 | 5.67 |
|---|---|---|---|---|---|---|


NN and XGBoost produced data distributions that were similar to SHARP (Fig. 7). SLR had the worst performance.
Fig. 7 shows that SLR could not predict low concentrations. The predictions made by NN and XGBoost ranged from
0.19 µg m$^{-3}$ to 47.19 µg m$^{-3}$ and from 0.00 µg m$^{-3}$ to 39.94 µg m$^{-3}$.

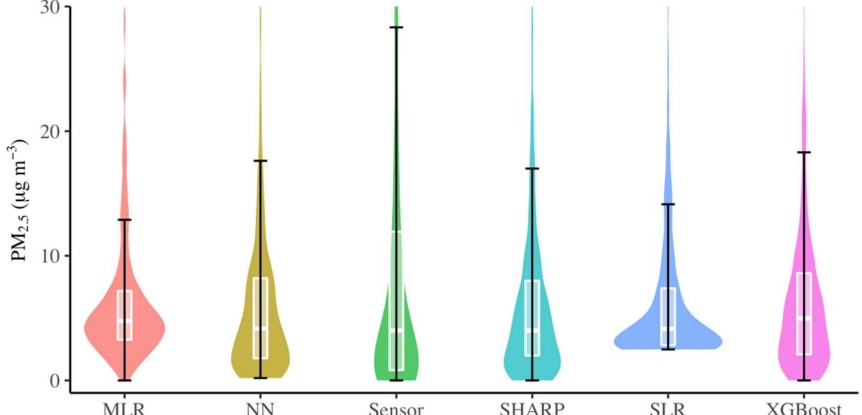


**Figure 7**: Data Density Comparison in the Test Dataset. Based on 610 Test Examples. NN: neural network, MRL: Multiple Linear
Regression, SLR: Simple Linear Regression. PM$_{2.5}$ data greater than 30 µg m$^{-3}$ are not shown in the figure. See the boxplot explanation in
Figure 3.






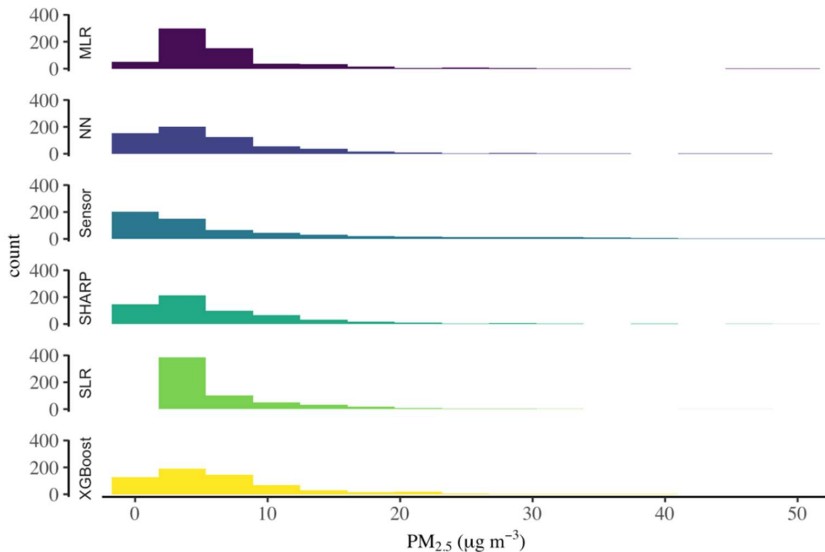

**Figure 8**: Data Distribution Comparison. Based on 610 Test Examples. NN: neural network, MRL: Multiple Linear Regression, SLR: Simple Linear Regression.

In the test dataset, the NN produced the lowest MAE of 2.38 (Fig. 9). The MAEs were 2.63 by XGBoost, 3.09 by MLR, and 3.21 by SLR, when compared with the $PM_{2.5}$ data measured by the SHARP instrument. The NN also had the lowest RMSE score in the test dataset. The RMSEs were 3.91 for the NN, 4.19 for XGBoost, and 9.93 for the low-cost sensor (Fig. 9). The Pearson r value by the NN was 0.85, compared to 0.74 by the low-cost sensor.



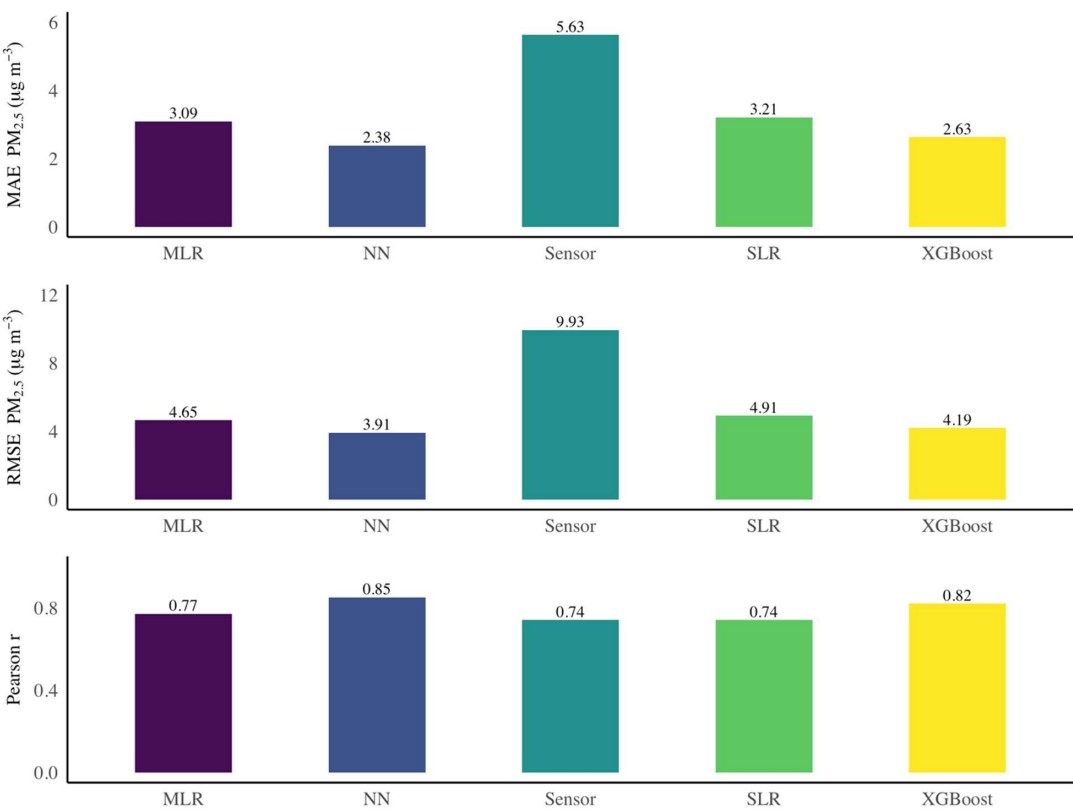

345

**Figure 9:** Performances of Different Calibration Methods. Based on 610 Test Examples. NN: neural network, MRL: Multiple Linear Regression, SLR: Simple Linear Regression.

The XGBoost and NN machine learning algorithms have a better performance, compared to traditional SLR and MRL calibration methods. NN calibration reduced RMSE by 60%. Both NN and XGBoost demonstrated the ability to correct the bias for high concentrations made by the low-cost sensor (Fig. 10 and Fig. 11). Most of the values that were greater than 10 $\mu$g m$^{-3}$ in the NN model fall closer to the yellow 1:1 line (Fig. 10). NN had slightly better performance for low concentrations compared to XGBoost.





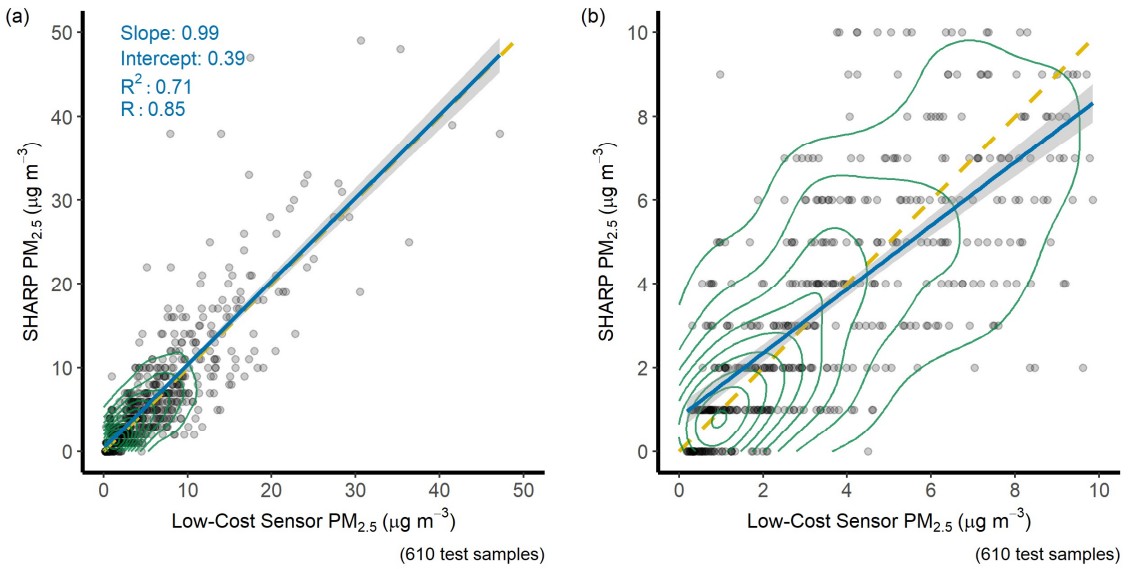

**Figure 10:** Comparison between the NN predictions and SHARP. Based on 610 test examples. Plot (a) is in full scale. Plot (b) is a zoom-in plot of plot (a). The solid blue line is a regression line. The yellow dashed line is a 1:1 line. The green circle represents data density. The grey area along the regression line represents 1 standard deviation.

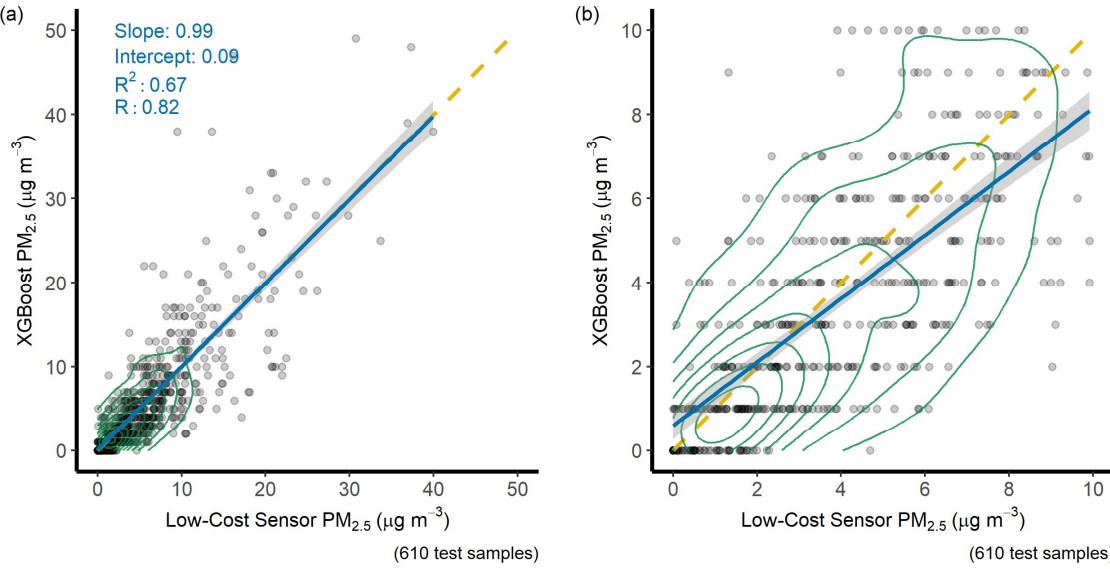



**Figure 11:** Comparison between the XGBoost predictions and SHARP. Based on 610 test examples. NN: Neural Network. Plot (a) is in full scale. Plot (b) is a zoom-in plot of plot (a). The solid blue line is a regression line. The yellow dashed line is a 1:1 line. The green circle represents data density. The grey area along the regression line represents 1 standard deviation.

## 4 Conclusions

In this study, we evaluated one low-cost sensor against a reference instrument – SHARP – using 3,050 hourly data from 00:00 on December 7, 2018, to 23:00 on April 26, 2019. The p-value from the F-K test suggested that the variances in the $PM_{2.5}$ values were statistically significantly different between the low-cost sensor and the SHARP instrument. Based on the 24-hour rolling average, the low-cost sensor in this study tended to report higher $PM_{2.5}$ values compared to the SHARP instrument. The low-cost sensor had strong bias when $PM_{2.5}$ concentrations were greater than 10 μg m$^{-3}$. The study also showed that the sensor's bias responses are likely caused by the sudden changes of RH.

Four calibration methods were tested and compared, including SLR, MLR, NN, and XGBoost. The p-values from the F-K tests for the XGBoost and NN were greater than 0.05, which indicated that, after calibration by the XGBoost and the NN, the variances of the $PM_{2.5}$ values were not statistically significantly different from the variance of the $PM_{2.5}$ values measured by the SHARP instrument. In contrast, the p-values from the F-K tests for the SLR and MLR were still less than 0.05. The NN generated the lowest RMSE score in the test dataset with 610 samples. The RMSE by NN was 3.91, the lowest of the four methods. RMSEs were 4.91 by SLP, 4.65 by MLR, and 4.19 by XGBoost.

However, a wide installation of low-cost sensors may still face challenges, including

- Durability of low-cost sensor. The low-cost sensor used in the study was deployed in ambient environment. We installed four sensors between December 7, 2018, and June 20, 2019. Only one sensor lasted approximately five months; the data from this sensor was used in this study. The other three sensors only lasted two weeks to one month and collected limited data. These three sensors did not collect enough data for machine learning and, therefore, were not used in this study.

- Missing data. In this study, the low-cost sensor dataset has 299 missing values for $PM_{2.5}$ concentrations. The reason for the missing data is unknown.

- Transferability of machine learning models. The models, developed by the two more powerful machine learning algorithms and used to calibrate the low-cost sensor data, tend to be sensor-specific because of the nature of machine learning. Further research is needed to test the transferability of the models for broader use.



*Data availability.* The hourly sensor data and hourly SHARP data are provided online at 10.5281/zenodo.3473833

*Author Contribution:* MS conducted evaluation and calibrations. YX installed the sensor and monitored and collected the
sensor data. MS and YX wrote the manuscript together and have equal contribution. SD edited the machine learning
methods. DK secured the funding and supervised the project. All authors discussed the results and commented on the
manuscript.

*Competing interests.* The authors declare no competing interest.

*Disclaimer.* Reference to any companies or specific commercial products does not constitute endorsement or
recommendation by the authors.

*Acknowledgments.* The authors wish to thank SensorUp for providing the low-cost sensors, and Calgary Region Airshed
Zone's air quality program manager Mandeep Dhaliwal for helping with the installation of the PM sensors and a 4G LTE
router, as well as the collection of the SHARP data. The authors would also like to thank Jessica Coles for editing this
manuscript.
The project was funded by Natural Sciences and Engineering Research Council of Canada (NSERC) Engage Program (No.
EGP 521823–17) and NSERC Collaborative Research and Development Program (No. CRDPJ 535813-18).

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
