# Peer review of "Evaluation and Calibration of a Low-cost Particle Sensor in Ambient"

_Atmospheric Measurement Techniques, 2019_

## Referee Comment (RC1) · Anonymous Referee #3 · 13 Jan 2020

Low-cost sensors are important components in environmental Internet of Things (IoT) for high resolution monitoring pollutant distribution over an area. This manuscript presents a method for addressing the challenge of low-cost PM sensor (i.e., questionable data quality) by using machine learning (ML) technologies. This study is interdisciplinary and is novel in terms of the application of random search techniques in ML configuration, as well as the application of ML on ambient PM monitoring over multiseasons. The results show ML has great potential for calibrating low-cost PM sensor for ambient aerosol monitoring. It would merit publication if the authors address/explain the follow comments/questions below.

1. Lines 147-150, the authors stated that random search is more efficient, which is also a unique part of this study. Please explain why it is more efficient than manual or grid search in principle and if possible, give some quantitative information.

2. Figure 1, I would appreciate a geographical map showing where is the monitoring location.

3. CRAZ also monitors NOx, NMHC, ozone, and wind data, which may also influence the PM concentration. Why these data were not included in the machine learning?

4. Line 195: there is a typo: "SHAPR" should be "SHARP".

5. SHARP was used as the reference method for PM monitoring. How often was SHARP calibrated to ensure its data quality.

6. Line 228: how the hyperparameters were determined?

7. Figure 3: will you explain what the shape of erlenmeyer flask means in the plot?

8. One aspect of the uniqueness of this study is that its study covers different seasons. I would like to see a brief discussion how season influence the results of low-cost sensors.

---

## Referee Comment (RC2) · Anonymous Referee #4 · 24 Feb 2020

This work built an IoT based low cost particle sensor equipment to collected long-term PM2.5 data to compare with a co-located US EPA FEM monitor, and then focused on using machine learning technologies to calibrate this low cost particle sensor.

The topic and method in this work is not new, many team did similar work using different sensors and calibration method. However, the team did some differentiate work comparing to others: 1. Use long-term data crossing 5 months to evaluate and calibrate the sensor over various RH and temperatures. 2. Low cost sensor is co-located with a US EPA FEM monitor, and the FEM monitor's data is used as the target to supervise the machine learning process 3. Evaluated several calibration methods including SLR,

MLR, XGBoost and NN with random searched hyperparameters.

The study result shows a convincing result that with feedforward NN calibration method, low cost sensor's test results and results of FEM monitor are not statistically significantly different. This may enable to build high spatial- and temporal-resolution PM monitoring networks to support public exposure and health effects studies that are related to PM.

There are several works could be addressed: 1. Work with sensor vendor to find out the reason of high equipment disability rate and build a larger sensor network in an area to evaluate the calibration method crossing different sensors. 2. Expand the temperature range to a warm condition, such as 30°C - 38°C to evaluation RH with higher temperature's effect on low cost sensor and calibration method.

---

## Author Comment (AC1) · 29 Feb 2020

1. **Comment 1 from Referee No. 3**: Lines 147-150, the authors stated that random search is more efficient, which is also a unique part of this study. Please explain why it is more efficient than manual or grid search in principle and if possible, give some quantitative information.

   **Author's response**: Manual search can be considered as an automated grid search. Grid search method and manual search method consider every combination of all the hyperparameters to build the learning models and each model needs to be evaluated to find out the one of the highest accuracy for training and

prediction. For the XGBoost algorithm used in the manuscript Section 2.3.2, we tuned 7 hyperparameters. Each hyperparameter has 20 different parameters. The grid is a 7 by 20 table. The complete grid search requires $20^7$ = 1,280 millions trials. In this study, we used 10 fold cross-validation, which means each trial will run 10 times. So the total runs will be 1,280 million $\times$ 10 = over 12 billion, which is computationally expensive.

For the random search, instead of computing the cases of all possible combinations, random combinations of hyperparameters are selected at each trial. Due to the random nature of sampling, the entire space of the grid could be reached (Zheng 2015).

The higher efficiency of random search can be explained by probability theory: Considering a sample space with a finite maximum, if we need to find a sample that is within the top 5% of all the samples, 60 random observations would give us 95% probability to find the sample. The value of 60 is calculated as follows:

As there are 5% eligible samples in the space, each random observation has 5% chance to find the eligible sample. On another hand, each random observation has $(1 - 5\%)$ chance not to find the eligible sample. If we take n random observations, the chance of not getting the eligible sample would be $(1 - 0.05)^n$, or the chance of getting the eligible sample would be $1 - (1 - 0.05)^n$. Let

$$1 - (1 - 0.05)^n > 95\%$$

And we can solve for n = 60 Therefore, the random search method would significantly save computation resources but still have a good chance to guess the close-to-optimal combination of hyperparameters.

**Author's changes in manuscript** We added a reference below in line 150 to explain the rationale of random search method. Zheng (2015) explained that random search with 60 samples will find a close-to-optimal combination with 95% of probability.

2. **Comment 2 from Referee No. 3** Figure 1, I would appreciate a geographical map showing where is the monitoring location.
**Author's response** a geographical map is added as Figure 2.
**Author's changes in manuscript**Added a new figure - Figure 2.

3. **Comment 3 from Referee No. 3** CRAZ also monitors NOx, NMHC, ozone, and wind data, which may also influence the PM concentration. Why these data were not included in the machine learning?
**Author's response** The low-cost sensor evaluated in this study only measured temperature (T) and relative humidity (RH). The ultimate goal of low-cost sensor application is to provide same quality data as the reference method using available information provided by the low-cost sensor. Therefore, we only used the parameters that the sensor measured. Next phase of the study would be testing other types of low-cost sensors, which may provide other parameters than T and RH. In that case, we would include those parameters in machine learning.
**Author's changes in manuscript** Not applicable.

4. **Comment 4 from Referee No. 3** Line 195: there is a typo: "SHAPR" should be "SHARP".
**Author's response** Corrected
**Author's changes in manuscript** Corrected to SHARP

5. **Comment 5 from Referee No. 3** SHARP was used as the reference method for PM monitoring. How often was SHARP calibrated to ensure its data quality
**Author's response** The SHARP instrument is regulated by the provincial air monitoring directive. It was calibrated monthly.
**Author's changes in manuscript** We added a clarification in Line 171 The instrument was calibrated monthly

6. **Comment 6 from Referee No. 3** Line 228: how the hyperparameters were determined?

**Author's response** The hyperparameters were determined by the XGBoost algorithm itself. Detailed explanation of each hyperparameter is provided in the XGBoost documentation https://xgboost.readthedocs.io/en/latest/parameter.html
**Author's changes in manuscript** We added the following reference in line 238 Detailed explanation of each hyperparameter is provided in the XGBoost documentation (XGBoost developers, 2019)

7. **Comment 7 from Referee No. 3** Figure 3: will you explain what the shape of erlenmeyer flask means in the plot?
   **Author's response** The plots outside of the boxplots in Figure 3 is called violin plot. The violin plot is to describe the density of data. More details can be found in the following link: https://mode.com/blog/violin-plot-examples/
   **Author's changes in manuscript** The following sentences were added in Line 264. The violin plot in Figure 3 describes the distribution of the PM2.5 values measured by the low-cost sensor and SHARP using density curve. The width of each curve represents the frequency of PM2.5 values at each concentration level.

8. **Comment 8 from Referee No. 3** One aspect of the uniqueness of this study is that its study covers different seasons. I would like to see a brief discussion how season influence the results of low-cost sensors.
   **Author's response** We added a section to discuss the seasonal impact in Section 3.5.2

   We assessed the seasonal impact on the low-cost sensor by comparing the mean of absolute daily average between the sensor values and the SHARP values in winter (December 2018 to February 2019) and spring (March 2019 to April 2019). A descriptive statistic is presented in Table 7.

   We used a two-sample t test to assess if the mean of absolute differences for winter and spring were statistically significant. The p value of the t test was

0.754. Because $P = 0.754 > \alpha = 0.05$, we retained the null hypothesis. There was not sufficient evidence at the $\alpha = 0.05$ level to conclude that the means of absolute differences between the low-cost sensor and SHARP PM values were significantly different for winter season and spring season.

**Author's changes in manuscript** Added a section 3.5.2

[Figure]

![Map of Canada with an inset satellite image showing the Varsity Air Monitoring Station near the University of Calgary Campus. The inset shows streets labeled 37 St NW, 33 St NW, 32 Ave NW, and Hwy 1A. The main map marks Calgary with a red star.]

**Fig. 1.**

---

## Author Comment (AC2) · 29 Feb 2020

1. **Comment 1 from Referee No. 4**: Work with sensor vendor to find out the reason of high equipment disability rate and build a larger sensor network in an area to evaluate the calibration method crossing different sensors

    **Author's response**: We thank the reviewers' recommendation for our future work. We think it might be caused by water damage to the controller board. We planned to carry out Phase 2 of the study to work with the sensor vendors or may try different sensors to understand what might be the cause that sensors do not last long. In phase 2, we also plan to deploy multiple sensors at multiple

Alberta air monitoring stations for a longer time, such as 1 year, so we can test sensor precision and bias, as well as transferability of machine learning models.

**Author's changes in manuscript** Not applicable

2. **Comment 2 from Referee No. 4** Expand the temperature range to a warm condition, such as 30 C- 38 C to evaluation RH with higher temperature's effect on low-cost sensor and calibration method

**Author's response** We thank the reviewers' recommendation for our future work. Because of sensor failure, the sensor we used did not last to summer. We plan to set up experiments in phase 2 to cover a wider range of weather conditions. We added a short discussion about season impacts as below.

We assessed the seasonal impact on the low-cost sensor by comparing the mean of absolute differences of the daily average between the sensor values and the SHARP values in winter (December 2018 to February 2019) and spring (March 2019 to April 2019). A descriptive statistic is presented in Table 7.

We used a two-sample t-test to assess if the mean of absolute differences for winter and spring were statistically significant. The p-value of the t-test was 0.754. Because $p = 0.754 > \alpha = 0.05$, we retained the null hypothesis. There was not sufficient evidence at the $\alpha = 0.05$ level to conclude that the means of absolute differences between the low-cost sensor and SHARP PM values were significantly different for the winter season and spring season.

**Author's changes in manuscript** Added a section 3.5.2

We assessed the seasonal impact on the low-cost sensor by comparing the mean of absolute differences of the daily average between the sensor values and the SHARP values in winter (December 2018 to February 2019) and spring (March 2019 to April 2019). A descriptive statistic is presented in Table 7.

We used a two-sample t-test to assess if the mean of absolute differences for winter and spring were statistically significant. The p-value of the t-test was 0.754.

Because $p = 0.754 > \alpha = 0.05$, we retained the null hypothesis. There was not sufficient evidence at the $\alpha = 0.05$ level to conclude that the means of absolute differences between the low-cost sensor and SHARP PM values were significantly different for the winter season and spring season.

---

## Author Response (AR1)

February 29, 2020

Dear AMT Editor:

**Re: Response to Reviewers' Comments from the Public Discussion - MS No.: amt-2019-393**

We thank the comments by the Referee No.3 and No.4 from the public discussion. Our responses are provided below.

| Comment 1 from Referee No. 3 | Lines 147-150, the authors stated that random search is more efficient, which is also a unique part of this study. Please explain why it is more efficient than manual or grid search in principle and if possible, give some quantitative information |
|---|---|
| Author's response | The grid search can be considered as an automated manual search. |
| | Grid search method and manual search method consider every combination of all the hyperparameters to build the learning models and each model needs to be evaluated to find out the one of the highest accuracies for training and prediction. For the XGBoost algorithm used in the manuscript Section 2.3.2, we tuned 7 hyperparameters. Each hyperparameter has 20 difference parameters. The grid looks like the following table: |

| Hyperparameter | P1 | … | P20 |
|---|---|---|---|
| H1 | | | |
| H2 | | | |
| … | | | |
| H7 | | | |

The complete grid search requires $20^7$ = 1,280 millions of trials. In this study, we used 10-fold cross validation, which means each trial will run 10 times. So, the total runs will be 1,280 millions * 10 = over 12 billion, which is computationally expensive.

For random search, instead of computing the cases of all possible combinations, random combinations of hyperparameters are selected at each trial. Due to the random nature of sampling, the entire space of the grid could be reached (Zheng 2015).

The higher efficiency of random search can be explained by probability theory: Considering a sample space with a finite

| | |
|---|---|
| | maximum, if we need to find a sample that is within the top 5% of all the samples, 60 random observations would give us 95% probability to find the sample. The value of 60 is calculated as follows: |
| | As there are 5% eligible samples in the space, each random observation has 5% of chance to find the eligible sample. On another hand, each random observation has (1-5%) chance not to find the eligible sample. If we take n random observations, the chance of not to get the eligible sample would be $(1-0.05)^n$, or the chance of getting the eligible sample would be $1-(1-0.05)^n$. Let |
| | $$1-(1-0.05)^n > 95\%$$ |
| | And we can solve for n = 60 |
| | Therefore, random search method would significantly save computation resource but still have a good chance to guess the close-to-optimal combination of hyperparameters. |
| Author's changes in manuscript | We added a reference below in line 150 to explain the rationale of random search method. |
| | Zheng (2015) explained that random search with 60 samples will find a close-to-optimal combination with 95% of probability. |

| | |
|---|---|
| Comment 2 from Referee No. 3 | Figure 1, I would appreciate a geographical map showing where is the monitoring location. |
| Author's response | a geographical map is added as Figure 2 |
| Author's changes in manuscript | Added a new figure - Figure 2. |

| | |
|---|---|
| Comment 3 from Referee No. 3 | CRAZ also monitors NOx, NMHC, ozone, and wind data, which may also influence the PM concentration. Why these data were not included in the machine learning? |
| Author's response | The low-cost sensor evaluated in this study only measured temperature (T) and relative humidity (RH). The ultimate goal of low-cost sensor application is to provide same quality data as the reference method using available information provided by the low-cost sensor. Therefore, we only used the parameters that the sensor measured. |
| | Next phase of the study would be testing other types of low-cost sensors, which may provide other parameters than T and RH. In that case, we would include those parameters in machine learning. |

| Author's changes in manuscript | Not applicable. |
|---|---|

| Comment 4 from Referee No. 3 | Line 195: there is a typo: "SHAPR" should be "SHARP". |
|---|---|
| Author's response | Corrected |
| Author's changes in manuscript | Corrected to
 SHARP |

| Comment 5 from Referee No. 3 | SHARP was used as the reference method for PM monitoring. How often was SHARP calibrated to ensure its data quality |
|---|---|
| Author's response | The SHARP instrument is regulated by the provincial air monitoring directive. It was calibrated monthly. |
| Author's changes in manuscript | We added a clarification in Line 171
 The instrument was calibrated monthly |

| Comment 6 from Referee No. 3 | Line 228: how the hyperparameters were determined? |
|---|---|
| Author's response | The hyperparameters were determined by the XGBoost algorithm itself.

 Detailed explanation of each hyperparameter is provided in the XGBoost documentation https://xgboost.readthedocs.io/en/latest/parameter.html |
| Author's changes in manuscript | We added the following reference in line 238
 Detailed explanation of each hyperparameter is provided in the XGBoost documentation (XGBoost developers, 2019) |

| Comment 7 from Referee No. 3 | Figure 3: will you explain what the shape of erlenmeyer flask means in the plot? |
|---|---|
| Author's response | The plots outside of the boxplots in Figure 3 is called violin plot. The violin plot is to describe the density of data. More details can be found in the following link: |

| | https://mode.com/blog/violin-plot-examples/ |
|---|---|
| Author's changes in manuscript | The following sentences were added in Line 264

The violin plot in Figure 3 describes the distribution of the PM2.5 values measured by the low-cost sensor and SHARP using density curve. The width of each curve represents the frequency of PM2.5 values at each concentration level. |

| | |
|---|---|
| Comment 8 from Referee No. 3 | One aspect of the uniqueness of this study is that its study covers different seasons. I would like to see a brief discussion how season influence the results of low-cost sensors. |
| Author's response | We added a section to discuss the seasonal impact in Section 3.5.2

We assessed the seasonal impact on the low-cost sensor by comparing the mean of absolute differences between the daily average of sensor values and the daily average of SHARP values in winter (December 2018 to February 2019) and spring (March 2019 to April 2019). A descriptive statistic is presented in Table 7.

We used a two-sample t test to assess if the means of absolute differences for winter and spring were equal. The p value of the t test was 0.754. Because P = 0.754 > α = 0.05, we retained the null hypothesis. There was not sufficient evidence at the α = 0.05 level to conclude that the means of absolute differences between the low-cost sensor and SHARP PM values were significantly different for winter season and spring season. |
| Author's changes in manuscript | Added a section 3.5.2 |

| | |
|---|---|
| Comment 1 from Referee No. 4 | Work with sensor vendor to find out the reason of high equipment disability rate and build a larger sensor network in an area to evaluate the calibration method crossing different sensors |
| Author's response | We thank the reviewers' recommendation for our future work. We think it might be caused by water damage to the controller board.

We planned to carry out a Phase 2 of the study to work with the sensor vendors or may try different sensors to understand what might be the cause that sensors do not last long.

In phase 2, we also plan to deploy multiple sensors at multiple Alberta air monitoring stations for a longer time, such as 1 year, so |

| | we can test sensor precision and bias, as well as transferability of machine learning models. |
|---|---|
| Author's changes in manuscript | Not applicable |

| | |
|---|---|
| Comment 2 from Referee No. 4 | Expand the temperature range to a warm condition, such as 30 C- 38 C to evaluation RH with higher temperature's effect on low cost sensor and calibration method |
| Author's response | We thank the reviewers' recommendation for our future work. |
| | Because of sensor failure, the sensor we used did not last to summer. We plan to set up experiments in phase 2 to cover a boarder ranges of weather condition |
| | We added a short discussion about season impacts. |
| | We assessed the seasonal impact on the low-cost sensor by comparing the mean of absolute differences between the daily average of sensor values and the daily average of SHARP values in winter (December 2018 to February 2019) and spring (March 2019 to April 2019). A descriptive statistic is presented in Table 7. |
| | We used a two-sample t test to assess if the means of absolute differences for winter and spring were equal. The p value of the t test was 0.754. Because $P = 0.754 > \alpha = 0.05$, we retained the null hypothesis. There was not sufficient evidence at the $\alpha = 0.05$ level to conclude that the means of absolute differences between the low-cost sensor and SHARP PM values were significantly different for winter season and spring season. |
| Author's changes in manuscript | Added a section 3.5.2 |

Respectful submitted

Calgary, Alberta Canada

Si et al.

[revised manuscript text omitted]

**3.5.2 Seasonal impact**

We assessed the seasonal impact on the low-cost sensor by comparing the mean of absolute differences of daily average between the sensor values and the SHARP values in winter (December 2018 to February 2019) and spring (March 2019 to

April 2019). A descriptive statistics is presented in Table 7.

**Table 7: Descriptive Statistics by Seasons**

| Season | Sample Size (n) | Mean[1] | Standard Deviation |
|--------|-----------------|---------|--------------------|
| Winter | 78 | 5.13 | 6.95 |
| Spring | 57 | 4.76 | 6.45 |

Note: 1) Mean is calculated by $\sum_{i=1}^{n}(|\ (sensor_{daily_i} - SHARP_{daily_i})|)/n$

We used a two-sample t test to assess if the average differences for winter and spring were statistically significant. The p value of the t test was 0.754. Because P = 0.754 > α = 0.05, we retained the null hypothesis. There was not sufficient evidence at the α = 0.05 level to conclude that the means of absolute differences between the low-cost sensor and SHARP

PM values were siginicantly different for winter season and spring season.

[revised manuscript text omitted]

de Winter, J. C. F., Gosling, S. D. and Potter, J.: Comparing the Pearson and Spearman correlation coefficients across
distributions and sample sizes: A tutorial using simulations and empirical data., Psychol. Methods, 21(3), 273–290,
doi:10.1037/met0000079, 2016.

XGBoost developers: XGBoost Parameters — xgboost 1.0.0-SNAPSHOT documentation, [online] Available from:
https://xgboost.readthedocs.io/en/latest/parameter.html (Accessed 24 January 2020), 2019.

Xiong, Y., Zhou, J., Schauer, J. J., Yu, W. and Hu, Y.: Seasonal and spatial differences in source contributions to PM2.5 in
Wuhan, China, Sci. Total Environ., 577, 155–165, doi:10.1016/j.scitotenv.2016.10.150, 2017.

Zheng, A.: Evaluating Machine Learning Models, First Edition., O'Reilly Media, Inc., 1005 Gravenstein Highway North,
Sebastopol, CA., 2015.

[revised manuscript text omitted]